# Death and invasive mechanical ventilation risk in hospitalized COVID-19 patients treated with anti-SARS-CoV-2 monoclonal antibodies and/ or antiviral agents: A systematic review and network meta-analysis protocol

Sumanta Saha *

Department of Community Medicine, R. G. Kar Medical College, Kolkata, West Bengal, India

* sumanta.saha@uq.net.au

## Abstract

### Background

The ongoing COVID-19 pandemic has claimed >4 million lives globally, and these deaths often occurred in hospitalized patients with comorbidities. Therefore, the proposed review aims to distinguish the inpatient mortality and invasive mechanical ventilation risk in COVID-19 patients treated with the anti-SARS-CoV-2 monoclonal antibodies and/or the antiviral agents.

### Methods

A search in PubMed, Embase, and Scopus will ensue for the publications on randomized controlled trials testing the above, irrespective of the publication date or geographic boundary. Risk of bias assessment of the studies included in the review will occur using the Cochrane risk of bias tool for randomized trials (RoB 2). Frequentist method network meta-analyses (NMA) will compare each outcome's risk across both types of anti-SARS-CoV-2 agents in one model and each in separate models. Additional NMA models will compare these in COVID-19 patients who were severely or critically ill, immunocompromised, admitted to the intensive care unit, diagnosed by nucleic acid amplification test, not treated with steroids, <18 years old, and at risk of infection due to variants of concern. The plan of excluding non-hospitalized patients from the proposed review is to minimize intransitivity risk. The acceptance of the network consistency assumption will transpire if the local and overall inconsistency assessment indicates no inconsistency. For each NMA model, the effect sizes (risk ratio) and their 95% confidence intervals will get reported in league tables. The best intervention prediction and quality of evidence grading will happen using the surface under the cumulative ranking curve values and the Grading of Recommendations Assessment, Development and Evaluation-based Confidence in Network Meta-Analysis approach, respectively. Sensitivity analysis will repeat the preliminary NMA while excluding the trials at high risk of bias. The Stata statistical software (v16) will be used for analysis. The statistical significance will get determined at p<0.05 and 95% confidence interval.

**Data Availability Statement:** Since this is systematic review protocol, no datasets were generated or analysed during the current study.

**Funding:** The author received no specific funding for this work.

**Competing interests:** The authors have declared that no competing interests exist.

## Trial registration

PROSPERO Registration No: https://www.crd.york.ac.uk/prospero/display_record.php?ID=CRD42021277663.

## Introduction

The ongoing COVID-19 pandemic has claimed more than 6 million lives globally as of 25-May-2022 [1]. COVID-19 occurs through exposure to individuals infected by SARS-CoV-2, an RNA virus with up to 14 days incubation period [2,3]. Existing meta-analytic estimates suggest a high mortality burden (nearly 17%) among hospitalized COVID-19 patients [4]. It's especially high in individuals with comorbid conditions like diabetes (about 20%) [5]. Severe disease and hospitalization risk in COVID-19 patients are associated with increased age, multiple pre-existing comorbidities, and poor management of these comorbid conditions [6]. COVID-19 patients with moderate, severe, or critical disease categories mainly require inpatient treatment [2]. The moderate disease category manifests with clinical or radiological signs of lower respiratory airway disease and sea-level room air oxygen saturation (SpO2) ≥94% [2]. The features of the severe form include sea-level room air SpO2 <94%, arterial partial pressure of oxygen to fraction of inspired oxygen ratio <300 mm Hg, respiratory rate >30 breaths/min, or lung infiltrates >50% [2]. The critical stage manifests with respiratory failure, septic shock, and/or multi-organ dysfunction [2]. Researches on therapies to decrease mortality in hospitalized COVID-19 patients are ongoing. Despite the ongoing mass vaccination drive, these researches are crucial as SARS-CoV-2's seizure capability of COVID-19 vaccinated or previous COVID-19 infected persons' immunity is increasing due to the continuous evolvement of the virus through random mutation [2]. Furthermore, transmissibility, virulence, and susceptibility to vaccines and therapeutics-related information are still evolving for World Health Organization declared variants of concern (e.g., B.1.617.2 (Delta) variant) [2].

Hospitalized COVID-19 patient mortality has been heavily studied in clinical trials using anti-SARS-CoV-2 antiviral agents and monoclonal antibodies. The common antiviral agents tested against SARS-CoV-2 include remdesivir, ivermectin, nitazoxanide, hydroxychloroquine, chloroquine, azithromycin, and human immunodeficiency virus protease inhibitors (e.g., lopinavir) [2]. Antiviral agents' mechanism of action against SARS-CoV-2 mainly includes viral entry inhibition via blockage of angiotensin-converting enzyme 2 receptor and transmembrane serine protease 2 or inhibition of fusion and endocytosis of the viral membrane and inhibition of enzymatic action of RNA-dependent RNA polymerase [7]. Presently, remdesivir is the only Food and Drug Administration-approved antiviral drug for the treatment of COVID-19 [2]; however, existing double-blinded randomized controlled trials (RCT) testing it haven't suggested any mortality benefit in hospitalized COVID-19 patients [8,9]. Likewise, the trials testing the effect of hydroxychloroquine and/ azithromycin [10–13] and HIV protease inhibitors didn't suggest any mortality benefit in hospitalized COVID-19 patients [13,14]. In this regard, there is insufficient evidence on ivermectin and nitazoxanide. Next, the spike protein of the SARS-CoV-2 genome targeting monoclonal antibodies has shown effectiveness in COVID-19 treatment [2]. Spike proteins enter the host cell by attaching themselves to the host angiotensin-converting enzyme 2 receptor [15]. Two large RCTs on tocilizumab (REMAP-CAP and RECOVERY trials) suggested mortality benefits in hospitalized COVID-19 patients [16,17]. However, the US COVID-19 Treatment Guidelines Panel (the Panel) currently recommends anti-SARS-CoV-2 monoclonal antibodies for

nonhospitalized mild to moderate COVID-19 patients at an increased risk of developing severe disease [2].

Given the increasing number of trials testing these anti-SARS-CoV-2 antiviral agents and monoclonal antibodies in hospitalized COVID-19 patients, clinicians treating them will be interested in the all-cause inpatient mortality risk variation across these agents to make an evidence-based treatment choice. While a few network meta-analyses (NMA) have reviewed the topic, these didn't entirely focus on hospitalized anti-SARS-CoV-2 antiviral agents and/or anti-SARS-CoV-2 monoclonal antibodies treated confirmed COVID-19 cases [18–20]. Henceforth, this systematic review and NMA protocol aims to compare the all-cause inpatient mortality risk among hospitalized COVID-19 cases treated with these drugs. It will additionally compare their risk of post-hospitalization invasive mechanical ventilation requirements.

## Method

The proposed review is registered in PROSPERO (registration no. CRD42021277663) [21]. The reporting of this report follows the Preferred Reporting Items for Systematic Review and Meta-Analysis Protocols (PRISMA-P) (2015) reporting system (S1 File) [22].

### Eligibility criteria

*Inclusion criteria*:

1. Study design: Parallel arm RCTs with any number of intervention arms.

2. Study population: COVID-19 diagnosed patients hospitalized for COVID-19 management.

3. Intervention group: The intervention arm/s participants must receive monoclonal antibody and/or antiviral therapy against SARS-CoV-2 with standard COVID-19 care.

4. Comparator group: The comparator arm should receive standard COVID-19 care with or without a placebo.

5. Outcome: The primary outcome of interest will be all-cause inpatient deaths. The secondary outcome will include eventual invasive mechanical ventilation requirements in hospitalized COVID-19 patients. A trial must report either one or both of these outcomes to be eligible.

COVID-19 diagnosis, disease severity definition, standard care regimen, and tested drugs' dosage and regimen will get accepted as per the trialists.

*Exclusion criteria*:

1. Clinical trials of other designs. E.g., single-arm trial.

2. COVID-19 patients managed in ambulatory settings or home care.

3. Trials testing COVID-19 vaccines.

4. Trial with probable or suspected cases of COVID-19 (trialist defined).

### Information sources and search strategy

A search for articles (published in any language) matching the above-stated eligibility criteria will ensue in the PubMed, Embase, and Scopus databases, not limited to any publication date and geographic boundary.

The database search will follow the following steps:

1. Theme generation based on key concepts of the research question.

2. Formation of search strings based on the theme.

3. Ascertaining appropriateness of a search string by finding four pre-identified eligible publications from respective databases [23]. Such search strings will be used to retrieve citations.

Following is a demonstration of the above method done in the PubMed database. Using the formulated themes 'COVID-19' and 'RCT,' subsequent search strings are proposed:

1. ("COVID-19"[MeSH Terms] NOT "COVID-19 Vaccines"[MeSH Terms]) AND (randomizedcontrolledtrial[Filter])

2. (("SARS-CoV-2"[Title/Abstract] OR "covid 19"[Title/Abstract] OR "covid 19"[Title/Abstract] OR "coronavirus"[Title/Abstract]) NOT "vaccine*"[Title/Abstract]) AND (randomizedcontrolledtrial[Filter])

The search strings above included the vaccine-related search terms after the Boolean operator 'NOT' to focus the search on the therapeutic RCTs and not on COVID-19-vaccine-related trials. The second search string used words and phrases instead of the MeSH terms as their manual tagging with the newly indexed articles in the PubMed database requires some time [24]. The above search strings were considered appropriate as these retrieved the four pre-identified articles indexed in the PubMed database [25–28]. On relevancy-wise sorting, the articles ranked within the top 50 citations (S2 File).

Supplementary searches will ensue in the bibliography of the publications included in the proposed review and unpublished literature (e.g., preprint servers like medRxiv) [29].

## Study selection

After uploading the database search retrieved citations in a referencing software, the study selection process will follow the successive steps:

1. Elimination of duplicate articles.

2. Skimming the title and abstract of the remaining citations while matching them against the above-stated eligibility criteria.

3. Full-text reading of publications appearing dubious or eligible for inclusion in the proposed review.

4. Finalizing articles to be reviewed and enlisting (with reason) articles read in full text.

## Data abstraction

The following data will get abstracted from the articles included in the proposed review in pre-piloted forms:

1. Study: Trial design, trial id, trial duration, the country of conduct, ethical clearance, funding, and participant consent.

2. Participant: Number of participants randomized to respective treatment arms and demographic characteristics of the participants in each intervention arm (e.g., gender distribution, mean age).

3. Intervention: The treatment given to each intervention arm with dose and regimen.

4. Outcome: Data on deaths and invasive mechanical ventilation.

### Risk of bias (RoB) in individual studies

Utilizing the Revised Cochrane RoB tool for randomized trials (RoB 2), the different RoB domains (bias originating from randomization, deviation from the intended intervention, method of outcome measurement, and reporting of results) will get assessed by answering signaling questions described elsewhere [30] as yes, probably yes, probably no, no, and no information [30]. Finally, an overall assessment will transpire across the domain-specific judgments [30].

### Role of review authors

Two or more review authors will conduct the database search, study selection, data abstraction, and RoB assessment. They will discourse to resolve any conflict in judgment and seek a third-party opinion (including contacting the trialists) on failing to achieve resolution.

### Data synthesis

**NMA.**    Using the frequentist method NMA the risk of an outcome between two interventions will be compared. Since a decreased occurrence of these outcomes is favorable and desired, a reduction in effect size will depict relative safety.

For respective outcomes of interest, the following NMA models are planned:

1. NMA across all interventions of interest (NMA1).

2. NMA across anti-SARS-CoV-2 monoclonal antibodies (NMA2).

3. NMA across anti-SARS-CoV-2 antiviral agents (NMA3).

The common comparator in the above models will comprise standard care recipients not receiving the drugs planned for testing in the proposed review.

An NMA will be performed for NMA models meeting the following criteria [31,32].

1. Low risk of heterogeneity: A pairwise meta-analysis (PMA) will precede each NMA to assess heterogeneity across the trials planned for inclusion in the NMA model. This heterogeneity assessment will require at least 20 studies for PMA, and/or the average sample size $\geq$80 to ensure an adequately powered heterogeneity assessment (80%) [33]. Due to the pandemic nature of the disease, trials across the globe are anticipated, which are unlikely to be similar in characteristics and setting. Therefore, the heterogeneity evaluation will happen using random-effect PMA (inverse variance method). A value of 0.5 will be added to each of the cells of the 2x2 table if there are zero events in any of the intervention arms in the PMA models. The statistical evaluation of heterogeneity will happen using $Chi^2$ (p<0.1 denoting the presence of heterogeneity) [34] and $I^2$ statistics (to quantifying heterogeneity; values of 25, 50, and 75% representing heterogeneity as low, moderate, and high, respectively) [35]. NMA will ensue if an adequately powered heterogeneity assessment suggests a low risk of heterogeneity, i.e., $I^2 \leq$25% and $Chi^2$ statistics p<0.1.

2. Network type: A connected network is required.

3. Degree of freedom for heterogeneity in the network: Should be present to allow random-effect consistency model fitting.

4. Degree of freedom for inconsistency in a network: Should be present to allow an inconsistency model fitting.

**Network map.** Network maps will get constructed to visually assess the relationship across interventions included in the NMA models. The nodes and their connectors in the network map will depict the interventions and the trials testing these interventions, respectively. The greater the number of trials comparing two interventions the thicker these connectors will be. Excessive overlapping of lines-led visually intricate network maps will be simplified by a repetitive treatment pair swapping [36].

**Transitivity and consistency.** The study population of the proposed review will not include non-hospitalized COVID-19 patients (as pre-stated in the eligibility criteria) to minimize intransitivity risk. Besides, a statistical evaluation of transitivity will comprise local (node-splitting method for testing inconsistency in each of the treatment pairs) and overall inconsistency assessment. A network consistency assumption will get accepted if both tests indicate an absence of inconsistency.

**Handling of intervention arms in NMA models.** The inclusion method of drug types tested in multi-arm trials in NMA models are stated below-

1. If respective treatment arms of an RCT tested a drug of interest alone and in combination with another drug of interest, their inclusion in the NMA model will happen as distinct interventions. E.g., outcomes across intervention arms testing the following in respective treatment arms of a hypothetical trial will not get clubbed- tocilizumab, remdesivir, and a combination of these two.

2. If the outcome data comes from various intervention arms of a trial that tested different dosages of the same drug, the clubbed data across such groups will get incorporated into the NMA model.

In all NMA models, the trial arms receiving standard COVID-19 care with or without a placebo that didn't receive the tested intervention of interest will form the common comparator.

**Handling of zero events.** An augmentation method will be used for the inclusion of trials with zero events in the NMA models. It will add a small amount of data (a value of 0.5) to all intervention arms.

**NMA effect sizes and ranking probabilities.** The league tables will present the effect size (in risk ratio) and their 95% CI comparing the risk of an outcome between two interventions. The diagonal cells of these tables will represent the interventions included in the NMA model.

When the league table of an outcome suggests at least one statistically significant favorable effect, the safest intervention for that outcome will be determined utilizing the surface under the cumulative ranking curve values. These values range between 0–100%, and higher values indicate better ranking, suggesting a safer intervention.

**Risk of bias across studies.** The publication bias evaluation will transpire utilizing the comparison-adjusted funnel plots as the clinical trials included in the proposed review will have a comparator treatment arm not receiving the tested interventions [37,38].

**Supplementary analysis.** NMA1, 2, and 3 will be carried out for each of the following COVID-19 patient cohorts:

1. Severe and critical patients.

2. Patients recruited in trials after December 2020, since when World Health Organization declared the emergence of SARS-CoV-2 variants of concern [2].

3. ICU admitted patients.

4. Immunocompromised patients. E.g., malignancy.

5. Nucleic acid amplification test diagnosed patients.

6. Steroids were not used as a part of the standard care regimen.

7. <18 years old patients.

**Sensitivity analysis.**   The sensitivity analysis will repeat NMA1, 2, and 3 for each outcome by eliminating trial/s at high risk of bias.

**Analytic tools.**   PMA and NMA will transpire using the 'meta' and 'network' packages of Stata statistical software version 16.0 (StataCorp, College Station, Texas, USA). Statistical significance estimation will happen at p<0.05 and 95% CI.

## Reporting of the review

The proposed review's reporting will adhere to PRISMA for Network Meta-Analyses statement guidelines [39].

## Confidence in cumulative evidence

The evidence quality assessment will happen in six domains (within-study bias, reporting bias, indirectness, imprecision, heterogeneity, and incoherence) using the Confidence in Network Meta-Analysis (CINeMA) approach [40]. The judgment in respective domains will happen at three levels- no, some, or major concerns [40]. Finally, an overall assessment across these domains will categorize the confidence level into very low, low, moderate, or high [40].

## Ethics and dissemination

Since this article is a protocol for a prospective systematic review and NMA, an ethical clearance requirement doesn't apply. The dissemination of the completed review will happen through a conference presentation and/or publication in a journal.

## Supporting information

**S1 File. PRISMA checklist.** Preferred Reporting Items for Systematic review and Meta-Analysis Protocols (PRISMA-P) 2015 checklist.
(DOCX)

**S2 File. Proposed search strategy.**
(DOCX)

## Author Contributions

**Conceptualization:** Sumanta Saha.

**Methodology:** Sumanta Saha.

**Writing – original draft:** Sumanta Saha.

**Writing – review & editing:** Sumanta Saha.

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
