## [Decision Letter · Decision Letter 0]

18 May 2022

PONE-D-21-31705

Death and invasive mechanical ventilation risk in anti-SARS-CoV-2 monoclonal antibodies and antiviral agents treated hospitalized COVID-19 patients: A systematic review and network meta-analysis protocol

PLOS ONE

Dear Dr. Saha,

Thank you for submitting your manuscript to PLOS ONE. After careful consideration, we feel that it has merit but does not fully meet PLOS ONE’s publication criteria as it currently stands. Therefore, we invite you to submit a revised version of the manuscript that addresses the points raised during the review process.

Please revise.

We look forward to receiving your revised manuscript.

Kind regards,

Academic Editor

PLOS ONE

Journal Requirements:

4. We note that this manuscript is a systematic review or meta-analysis; our author guidelines therefore require that you use PRISMA guidance to help improve reporting quality of this type of study. Please upload copies of the completed PRISMA checklist as Supporting Information with a file name “PRISMA checklist”.

Reviewers' comments:

Reviewer's Responses to Questions

**Comments to the Author**

1. Does the manuscript provide a valid rationale for the proposed study, with clearly identified and justified research questions?

Reviewer #1: Partly

Reviewer #2: Yes

2. Is the protocol technically sound and planned in a manner that will lead to a meaningful outcome and allow testing the stated hypotheses?

Reviewer #1: Partly

Reviewer #2: Yes

3. Is the methodology feasible and described in sufficient detail to allow the work to be replicable?

Reviewer #1: Yes

Reviewer #2: Yes

4. Have the authors described where all data underlying the findings will be made available when the study is complete?

Reviewer #1: No

Reviewer #2: Yes

5. Is the manuscript presented in an intelligible fashion and written in standard English?

Reviewer #1: Yes

Reviewer #2: Yes

6. Review Comments to the Author

You may also provide optional suggestions and comments to authors that they might find helpful in planning their study.

Reviewer #1: Peer Reviewer description

The following manuscript was submitted to the PLOS ONE journal “Death and invasive mechanical ventilation risk in anti-SARS-CoV-2 monoclonal antibodies and antiviral agents treated hospitalized COVID-19 patients: A systematic review and network meta-analysis protocol”.

The purpose of the following systematic review protocol is to summarize evidence from RCTs, regarding the effectiveness of antiviral agents and/or monoclonal antibodies on the mortality and invasive mechanical ventilation risk of hospitalized patients diagnosed with Covid-19.

Other studies have used a Network meta-analysis approach to assess the effectiveness of Covid-19 drugs (For example: Siemieniuk 2020). This review includes some of the drugs´ targeted by the authors. It would be important to cite this study and to mention what would be added or what would be the innovation proposed by the authors.

The authors provided a detailed rationale (based on the PRISMA suggestions) about the methods they will employ in their systematic review. Here are some suggestions:

Title of the study could be improved, for example: Death and invasive mechanical ventilation risk in hospitalized COVID-19 patients treated with anti-SARS-CoV-2 monoclonal antibodies and/or antiviral agents.

A strength of the systematic review process is to assess all the information available. The authors are proposing to only include articles in English, if so, please provide a rationale to justify this decision. Also, there is no mention of including grey literature or other sources of information (unpublished data, contact with experts, searching reference list).

How co-interventions would be addressed? For example, studies in which steroids are part of the standard of care, compared to studies in which is not.

How the pre-state notions will be addressed? Are the authors using a specific tool to assess the selective reporting (ORBIT tool)? Are the authors considering to assess publication bias? Please elaborate more about this section.

The authors mentioned they will be using the GRADE framework to assess the confidence of the cumulative evidence. The authors cited the pairwise approach of the GRADE framework. The GRADE framework for network meta-analysis has a different approach than the overall GRADE framework. The direct estimates are evaluated with the common approach, the indirect estimate is evaluated with the intransitivity and imprecision domains and the network estimate are evaluated with the imprecision domain. If authors are aiming to use the GRADE approach, please elaborate more about this section.

References

Siemieniuk R A, Bartoszko J J, Ge L, Zeraatkar D, Izcovich A, Kum E et al. Drug treatments for covid-19: living systematic review and network meta-analysis BMJ 2020; 370: m2980 doi:10.1136/bmj.m2980

Brignardello-Petersen R, Bonner A, Alexander PE, Siemieniuk RA, Furukawa TA, Rochwerg B, Hazlewood GS, Alhazzani W, Mustafa RA, Murad MH, Puhan MA, Schünemann HJ, Guyatt GH; GRADE Working Group. Advances in the GRADE approach to rate the certainty in estimates from a network meta-analysis. J Clin Epidemiol. 2018 Jan;93:36-44. doi: 10.1016/j.jclinepi.2017.10.005. Epub 2017 Oct 17. Erratum in: J Clin Epidemiol. 2018 Jun;98 :162. PMID: 29051107.

Brignardello-Petersen R, Mustafa RA, Siemieniuk RAC, Murad MH, Agoritsas T, Izcovich A, Schünemann HJ, Guyatt GH; GRADE Working Group. GRADE approach to rate the certainty from a network meta-analysis: addressing incoherence. J Clin Epidemiol. 2019 Apr;108:77-85. doi: 10.1016/j.jclinepi.2018.11.025. Epub 2018 Dec 5. PMID: 30529648.

Reviewer #2: This is a proposed protocol aiming to distinguish the inpatient mortality and invasive mechanical ventilation risk in COVID-19 patients treated with the anti-SARS-CoV-2 monoclonal antibodies and/or the antiviral agents. I have reviewed the document and have no comments.

7. PLOS authors have the option to publish the peer review history of their article (what does this mean?). If published, this will include your full peer review and any attached files.

Reviewer #1: No

Reviewer #2: No

---

## [Author Response · Author response to Decision Letter 0]

26 May 2022

Dear Reviewers,

 Thank you for reviewing this manuscript and sharing your feedback. The revised manuscript's amendments transpired according to your feedback. Moreover, a thorough revision of the entire manuscript ensued for a concise and coherent presentation of facts. Also, given that substantial time has passed since its submission to the journal (October 2021), relevant facts or data were checked for changes and incorporated as appropriate. For instance, the global COVID-19 death toll got upgraded to that of May 2022. 

Below are the replies to your comments.

Reviewer #1: 

Reviewer description

The following manuscript was submitted to the PLOS ONE journal “Death and invasive mechanical ventilation risk in anti-SARS-CoV-2 monoclonal antibodies and antiviral agents treated hospitalized COVID-19 patients: A systematic review and network meta-analysis protocol”.

The purpose of the following systematic review protocol is to summarize evidence from RCTs, regarding the effectiveness of antiviral agents and/or monoclonal antibodies on the mortality and invasive mechanical ventilation risk of hospitalized patients diagnosed with Covid-19.

REVIEWER COMMENT:

Other studies have used a Network meta-analysis approach to assess the effectiveness of Covid-19 drugs (For example: Siemieniuk 2020). This review includes some of the drugs´ targeted by the authors. It would be important to cite this study and to mention what would be added or what would be the innovation proposed by the authors. The authors provided a detailed rationale (based on the PRISMA suggestions) about the methods they will employ in their systematic review.

Author reply:

Thank you for the comment. Now the references to other network meta-analysis papers that reviewed COVID-19 drugs have been included, and the sentence in the updated manuscript reads as the following- ‘While a few network meta-analyses (NMA) have reviewed the topic, these didn't entirely focus on hospitalized anti-SARS-CoV-2 antiviral agents and/or anti-SARS-CoV-2 monoclonal antibodies treated confirmed COVID-19 cases [18–20].’

Following publications were cited for the purpose:

18. Siemieniuk RA, Bartoszko JJ, Ge L, Zeraatkar D, Izcovich A, Kum E, et al. Drug treatments for covid-19: living systematic review and network meta-analysis. BMJ [Internet]. 2020;m2980. Available from: https://www.bmj.com/lookup/doi/10.1136/bmj.m2980

19. Cheng Q, Chen J, Jia Q, Fang Z, Zhao G. Efficacy and safety of current medications for treating severe and non-severe COVID-19 patients: an updated network meta-analysis of randomized placebo-controlled trials. Aging (Albany NY) [Internet]. 2021;13:21866–902. Available from: http://www.ncbi.nlm.nih.gov/pubmed/34531332

20. Kim MS, An MH, Kim WJ, Hwang T-H. Comparative efficacy and safety of pharmacological interventions for the treatment of COVID-19: A systematic review and network meta-analysis. PLoS Med [Internet]. 2020;17:e1003501. Available from: http://www.ncbi.nlm.nih.gov/pubmed/33378357

REVIEWER COMMENT:

Here are some suggestions:

Title of the study could be improved, for example: Death and invasive mechanical ventilation risk in hospitalized COVID-19 patients treated with anti-SARS-CoV-2 monoclonal antibodies and/or antiviral agents.

Author reply:

Thank you for suggesting a more concise title. The updated title reads as the following- ‘Death and invasive mechanical ventilation risk in hospitalized COVID-19 patients treated with anti-SARS-CoV-2 monoclonal antibodies and/or antiviral agents: A systematic review and network meta-analysis protocol.’

REVIEWER COMMENT:

A strength of the systematic review process is to assess all the information available. The authors are proposing to only include articles in English, if so, please provide a rationale to justify this decision. Also, there is no mention of including grey literature or other sources of information (unpublished data, contact with experts, searching reference list).

Author reply:

Thanks for your comment. The language restriction to the English language and published literature only has been removed from the revised manuscript. 

The relevant sentences from the updated manuscript are as the following- 

‘A search for articles (published in any language) matching the above-stated eligibility criteria will ensue in the PubMed, Embase, and Scopus databases, not limited to any publication date and geographic boundary.’

‘Supplementary searches will ensue in the bibliography of the publications included in the proposed review and unpublished literature (e.g., preprint servers like medRxiv) [29].’

REVIEWER COMMENT:

How co-interventions would be addressed? For example, studies in which steroids are part of the standard of care, compared to studies in which is not.

Author reply:

Thanks for your comment. It is the tricky part indeed, and several steps are planned, therefore.

First, to distinguish the safety across different anti-SARS-CoV-2 drug categories of interest, three network meta-analyses (NMA) models (NMA 1, 2, and 3) are proposed. The NMA1 model will include data from trials testing anti-SARS-CoV-2 monoclonal antibodies and/or antivirals. The NMA2 and NMA3 models will incorporate the anti-SARS-CoV-2 monoclonal antibodies and the anti-SARS-CoV-2 monoclonal antivirals, respectively. 

Second, to address the relative safety of these interventions in different clinical scenarios, seven additional analyses have been proposed for each type of NMA model (under ‘Supplementary analysis’ subheading). It will, therefore, result in ≥21 (7x3=21) supplementary NMA models cross-examining the facts in different clinical-pharmacological scenarios.

Finally, to address your concern when steroids are part of standard care, in one of the supplementary analyses stated above, the NMA1, 2, and 3 models will include trials that didn't include steroids as a part of their standard care regimen. It will help distinguish how the outcomes vary in the presence or absence of steroids in standard care regimens. 

Relevant text from the revised manuscript is quoted here for your reference:

‘NMA1, 2, and 3 will be carried out for each of the following COVID-19 patient cohorts:

1. Severe and critical patients.

2. Patients recruited in trials after December 2020, since when World Health Organization declared the emergence of SARS-CoV-2 variants of concern [2].

3. ICU admitted patients.

4. Immunocompromised patients. E.g., malignancy.

5. Nucleic acid amplification test diagnosed patients.

6. Steroids were not used as a part of the standard care regimen.

7. <18 years old patients.’

REVIEWER COMMENT:

How the pre-state notions will be addressed? Are the authors using a specific tool to assess the selective reporting (ORBIT tool)? Are the authors considering to assess publication bias? Please elaborate more about this section.

Author reply:

Thank you for the comment. 

The pre-stated notion will get evaluated using the following domain of the Revised Cochrane risk-of-bias tool for randomized trials (RoB 2) tool- ‘bias in selection of the reported results.’ Since this is the most recently released critical appraisal tool, the previously stated JBI tool has been replaced with RoB 2 in the revised manuscript.

The risk of bias section has been amended in the following manner- ‘Utilizing the Revised Cochrane RoB tool for randomized trials (RoB 2), the different RoB domains (bias originating from randomization, deviation from the intended intervention, method of outcome measurement, and reporting of results) will get assessed by answering signaling questions described elsewhere [30] as yes, probably yes, probably no, no, and no information [30]. 

Finally, an overall assessment will transpire across the domain-specific judgments [30].’

Yes, publication bias evaluation will occur, and now it has been addressed in the manuscript in the following manner- ‘The publication bias evaluation will transpire utilizing the comparison-adjusted funnel plots as the clinical trials included in the proposed review will have a comparator treatment arm not receiving the tested interventions [37,38].’

REVIEWER COMMENT:

The authors mentioned they will be using the GRADE framework to assess the confidence of the cumulative evidence. The authors cited the pairwise approach of the GRADE framework. The GRADE framework for network meta-analysis has a different approach than the overall GRADE framework. The direct estimates are evaluated with the common approach, the indirect estimate is evaluated with the intransitivity and imprecision domains and the network estimate are evaluated with the imprecision domain. If authors are aiming to use the GRADE approach, please elaborate more about this section.

Author reply:

Thank you for your comment. Yes, the Grading of Recommendations Assessment, Development and Evaluation (GRADE)-based approach, the Confidence in Network Meta-Analysis (CINeMA) approach, will be used. 

The description in the manuscript has been updated and reads as the following- ‘The evidence quality assessment will happen in the six domains (within-study bias, reporting bias, indirectness, imprecision, heterogeneity, and incoherence) using the Confidence in Network Meta-Analysis (CINeMA) approach [40]. The judgment in respective domains will happen at three levels- no, some, or major concerns [40]. Finally, an overall assessment across these domains will categorize the confidence level into very low, low, moderate, or high [40].’

Thank you.

Reviewer #2: 

REVIEWER COMMENT:

This is a proposed protocol aiming to distinguish the inpatient mortality and invasive mechanical ventilation risk in COVID-19 patients treated with the anti-SARS-CoV-2 monoclonal antibodies and/or the antiviral agents. I have reviewed the document and have no comments.

Author reply:

Thank you.

---

## [Decision Letter · Decision Letter 1]

7 Jun 2022

Death and invasive mechanical ventilation risk in hospitalized COVID-19 patients treated with anti-SARS-CoV-2 monoclonal antibodies and/or antiviral agents: A systematic review and network meta-analysis protocol

PONE-D-21-31705R1

Dear Dr. Saha,

We’re pleased to inform you that your manuscript has been judged scientifically suitable for publication and will be formally accepted for publication once it meets all outstanding technical requirements.

Kind regards,

Academic Editor

PLOS ONE

Additional Editor Comments (optional):

Reviewers' comments:

Reviewer's Responses to Questions

**Comments to the Author**

1. Does the manuscript provide a valid rationale for the proposed study, with clearly identified and justified research questions?

Reviewer #1: Yes

Reviewer #2: Yes

2. Is the protocol technically sound and planned in a manner that will lead to a meaningful outcome and allow testing the stated hypotheses?

Reviewer #1: Yes

Reviewer #2: Yes

3. Is the methodology feasible and described in sufficient detail to allow the work to be replicable?

Reviewer #1: Yes

Reviewer #2: Yes

4. Have the authors described where all data underlying the findings will be made available when the study is complete?

Reviewer #1: Yes

Reviewer #2: Yes

5. Is the manuscript presented in an intelligible fashion and written in standard English?

Reviewer #1: Yes

Reviewer #2: Yes

6. Review Comments to the Author

You may also provide optional suggestions and comments to authors that they might find helpful in planning their study.

Reviewer #1: The authors have responded to all the previous comments, and they have provided justifications for their decisions.

Thanks for addressing the comments.

Reviewer #2: This is a proposed protocol aiming to find the mortality and invasive mechanical ventilation risk in hospitalized COVID-19 patients treated with anti-SARS-CoV-2 monoclonal antibodies and/or antiviral agents. I have reviewed the document and have no comments.

7. PLOS authors have the option to publish the peer review history of their article (what does this mean?). If published, this will include your full peer review and any attached files.

Reviewer #1: **Yes: **Luis Enrique Colunga Lozano

Reviewer #2: No

---

## [Editor Report · Acceptance letter]

10 Jun 2022

PONE-D-21-31705R1 

Death and invasive mechanical ventilation risk in hospitalized COVID-19 patients treated with anti-SARS-CoV-2 monoclonal antibodies and/or antiviral agents: A systematic review and network meta-analysis protocol 

Dear Dr. Saha:

I'm pleased to inform you that your manuscript has been deemed suitable for publication in PLOS ONE. Congratulations! Your manuscript is now with our production department. 

Kind regards, 

on behalf of

Dr. Robert Jeenchen Chen 

Academic Editor

PLOS ONE